# Preliminary Report of Nationwide COVID-19 Vaccine Compensation in Taiwan

**DOI:** 10.3390/healthcare12131250

**Published:** 2024-06-24

**Authors:** Yi-An Lu, Fu-Yuan Huang, Hsin Chi, Chien-Yu Lin, Nan-Chang Chiu

**Affiliations:** 1Division of Pediatric Infectious Diseases, Department of Pediatrics, Mackay Memorial Hospital, Taipei 10449, Taiwan; anne.4996@mmh.org.tw (Y.-A.L.); fyhuang@mmh.org.tw (F.-Y.H.); chi.4531@mmh.org.tw (H.C.); 2Department of Medicine, MacKay Medical College, New Taipei City 25245, Taiwan; 3Department of Pediatrics, Hsinchu Municipal MacKay Children’s Hospital, Hsinchu City 300, Taiwan

**Keywords:** coronavirus disease 2019 vaccine, vaccine adverse events, Vaccine Injury Compensation Program

## Abstract

The potential adverse effects of coronavirus disease 2019 (COVID-19) vaccinations raise public concerns. Data from Taiwan’s Vaccine Injury Compensation Program (VICP) can provide valuable insights. This study analyzed the preliminary application data for COVID-19 vaccine compensation in Taiwan’s VICP, focusing on applicants receiving vaccines between March 2021 and June 2022. Among the 2941 adverse events, 113 cases (3.8%) were deemed causally associated with vaccination, 313 (10.6%) were indeterminate, and 2515 (85.5%) had no causal association. Nearly half (47.6%) of the applicants were over 60 years old, and 76.6% had a history of pre-existing chronic diseases. Among the 426 vaccine-associated or indeterminate cases, the most common causes were hematological diseases and thrombosis. There were 920 mortality cases reported, and 97.4% were unassociated with vaccination. Only five deaths were judged to be associated with the COVID-19 vaccination, all involving the adenovirus vector vaccine and thrombosis with thrombocytopenia syndrome. In conclusion, most compensation applications were not causally linked to vaccination. Compared to other countries, the number of applications in Taiwan’s VICP is relatively high. These findings may indicate a need to adjust the application requirements for compensation in Taiwan’s program.

## 1. Introduction

The coronavirus disease 2019 (COVID-19) is a global epidemic characterized by severe outbreaks, and vaccination represents the most effective strategy to mitigate and ultimately end the pandemic. Various COVID-19 vaccines have received emergency use authorization to combat the infection and achieve widespread immunity across the entire population. Among these vaccines, one adenovirus vector COVID-19 vaccine ChAdOx1 nCoV-19 (AstraZeneca, AZ, Oxford, UK), two mRNA vaccines mRNA-1273 (Moderna, MDN, Cambridge, MA, USA) and BNT162b2 (Pfizer/BioNTech, BNT, New York, NY, USA), and one protein-based vaccine MVC-COV1901 (Medigen, MVC, Taipei, Taiwan) have received authorization for emergency use in Taiwan since February 2021 [1]. AZ was the first vaccine available in Taiwan, starting in March 2021 [1]. Subsequently, the MDN vaccine was introduced in June 2021, followed by the MVC vaccine in August 2021 and the BNT vaccine in September 2021 [2]. Billions of doses of COVID-19 vaccines have been administered worldwide in the fight against severe acute respiratory syndrome coronavirus 2. While many have lauded the efficiency of these newly developed vaccines, a considerable number of individuals remain skeptical. This skepticism has been exacerbated by a surge in reports on vaccine-related complications [3,4]. Vaccine refusal could be attributable to distrust in health authorities, low confidence in vaccines, misconceptions, and political beliefs [5]. In Taiwan, applications for the Vaccine Injury Compensation Program (VICP) are free and easy to access, leading to a relatively high number of applications compared to other countries. This study aimed to provide clinicians with an overview of the preliminary data regarding the demographics, clinical manifestations, and symptoms of COVID-19 vaccine Adverse Events Following Immunization (AEFI) reported to Taiwan’s VICP.

## 2. Materials and Methods

We compiled data on VICP judgments from January 2021 to June 2023 for COVID-19 vaccination compensation claims. For each patient, the following data were collected: vaccination types, vaccination dosage, age, gender, pre-existing chronic diseases, symptoms experienced, onset timing of symptoms, final diagnoses of symptoms after vaccination, prognosis, and VICP’s decision on the application. Pre-existing chronic diseases were categorized into cardiovascular diseases (such as hypertension, coronary artery diseases, cardiac arrhythmia, congenital cardiac malformation, etc.), endocrine/metabolic diseases (such as diabetes mellitus, hyperlipidemia, thyroid disorders, etc.), urinary system diseases (such as renal failure on dialysis, hydronephrosis, renal stones, renal tumors, etc.), neurological diseases (such as stroke, cerebral hemorrhage, epilepsy, dementia, Parkinsonism, psychiatric disorders, etc.), and others (such as hematological disorders, gastrointestinal and hepatobiliary disorders, chronic respiratory tract diseases, ophthalmic disorders, ear, nose, and throat disorders, dermatological disorders, orthopedic disorders, gynecological diseases, etc.). The compensation claims for diagnoses were classified into categories, such as cardiovascular diseases, neurological diseases, infectious diseases, hematological diseases, dermatological diseases, anaphylaxis, and others. Apart from anaphylaxis, the final diagnoses of symptoms following vaccination applications were grouped into five main categories: hematological, dermatological, neurological, cardiovascular diseases, and others. This study was approved by the Mackay Hospital Institutional Review Board (21MMHIS408e) and informed consent from participants was waived. This study was conducted following the Helsinki Declaration.

The VICP committee determined the association between adverse events and COVID-19 vaccines by reviewing patients’ medical charts and vaccination histories. Compensation was awarded based on the established association and the severity of the damage to health incurred. Based on VICP’s judgments, the association between adverse events and vaccines was categorized into three groups: causally associated, indeterminate, and unassociated. The VICP committee incorporated the WHO’s guideline “Causality assessment of an adverse event following immunization (AEFI): User manual for the revised WHO classification, 2019 update” when assessing the relationship between adverse events and vaccination.

Cases deemed associated or indeterminate required special attention, and they were grouped together for further analysis compared to the unassociated cases. For fatal cases, additional analyses were conducted, including the examination of autopsy results. Statistical analysis was performed using the Statistical Package for the Social Sciences (Version 26.0, SPSS, Inc., Chicago, IL, USA). The chi-squared test was used, as appropriate, to assess differences between categorical variables. A *p*-value of less than 0.05 was considered statistically significant.

## 3. Results

### 3.1. Demographics of the Applicants

From January 2021 to June 2023, 2941 cases of individuals inoculated with COVID-19 vaccines between March 2021 and June 2022 applied for the VICP. The distribution of cases by vaccine type was as follows: 1590 (54.1%) after the AZ vaccine, 782 (26.6%) after the MDN vaccine, 426 (14.5%) after the BNT vaccine, and 143 (4.9%) after the MVC vaccine (Appendix A). Among the applicants, 1547 (52.6%) were male and 1394 (47.4%) were female. The most common age group ranged from 40 to 79 years, with nearly half (47.6%) of cases being over 60 years old (Figure 1). Out of the total cases, 2202 individuals (74.8%) had complete medical records for the previous 3 years available in the VICP data. Pre-existing chronic diseases were identified in more than three-fourths (76.6%) of these individuals. Cardiovascular diseases were the most prevalent pre-existing condition, affecting 51.3% of individuals, followed by endocrine/metabolic diseases (40.2%), neurological diseases (30.0%), and urinary system diseases (11.9%). Comorbidities were common, with only 23.3% of cases having a single pre-existing disease, whereas the majority had multiple pre-existing chronic conditions.

### 3.2. Adverse Events following Vaccination

Approximately one-third (31.6%) of patients experiencing Adverse Event Following Immunization (AEFI) occurred within 1 day after the administration of the COVID-19 vaccine, with 42.9% occurring between 1 day and 1 week, 23.5% between 2 and 6 weeks, and 2.1% more than 6 weeks postvaccination (Appendix A). Figure 2 summarizes the diagnoses for compensation claims and the associated vaccine types. The most common claimed problems were related to the cardiovascular system, accounting for 24.7% (758 cases) of reported issues. This category included 379 cases (12.3%) of arrhythmia or heart failure, 286 cases (9.3%) of coronary artery disease or myocardial infarction, 55 cases (1.8%) of myocarditis or pericarditis, and 38 cases (1.2%) of aortic dissection. Manifestations related to the nervous system amounted to 691 cases (22.5%), including 477 cases (15.5%) of stroke, cerebral hemorrhage, or hypoxic brain lesions; 34 cases (1.1%) of Guillain–Barre syndrome (GBS); 12 cases (0.4%) of encephalitis; and 168 cases (5.5%) of peripheral nervous system disorders, such as facial palsy, hearing loss, or dizziness. Infection-related manifestations amounted to 389 cases (12.7%), including 260 cases (8.5%) of pneumonia or sepsis and 129 cases (4.2%) of local cellulitis. Thrombocytopenia and thrombosis were reported in 210 cases (6.8%), urticaria and other skin allergic reactions in 92 cases (3%), and anaphylaxis in 6 cases (0.2%).

### 3.3. Vaccine Types and Associations

Of the reported cases of AEFI, 113 cases (3.8%) were classified as causally associated with vaccination, 313 cases (10.6%) were indeterminate, and 2515 cases (85.5%) were deemed unassociated. Table 1 summarizes the demographic and clinical characteristics of the combined associated/indeterminate cases and unassociated cases. The combined associated/indeterminate group had more female and younger patients with fewer pre-existing chronic diseases and lower mortality rates but no significant differences in vaccine-symptom onset intervals. On the contrary, the unassociated group had higher MDN vaccination rates and lower AZ vaccination rates.

Among the 426 associated/indeterminate cases, the AZ vaccine accounted for 259 cases, with 91.9% (238 cases) following the first dose and 8.1% (21 cases) following the second dose. However, the MDN vaccine accounted for 82 cases, with 61.0% (50 cases) following the first dose, 35.4% (29 cases) following the second dose, and 3.6% (3 cases) following the third dose. Furthermore, the BNT vaccine accounted only for 65 cases, with 81.5% (53 cases) following the first dose, 17% (11 cases) following the second dose, and 1.5% (one case) following the third dose. Finally, the MVC vaccine accounted for 20 cases, with 70.0% (14 cases) following the first dose and 30.0% (6 cases) following the second dose.

Among the associated/indeterminate group, hematological diseases were the most common diagnoses (75 cases, 17.2%), followed by dermatological diseases (62 cases, 14.3%), neurological diseases (57 cases, 13.1%), and cardiovascular diseases (52 cases, 12%) (Figure 3, Appendix A). All cases of hematological diseases were inoculated with the AZ vaccine, with 94.7% occurring after the first dose. Twenty-five cases of thrombosis with thrombocytopenia syndrome (TTS) were diagnosed, five of which resulted in death. The time intervals between vaccine administration and symptom onset were between 1 and 15 days in these five patients. Most dermatological cases (58, 93.5%) were diagnosed as urticaria. GBS was reported in 43 cases, with 36 cases occurring after the AZ vaccination and 7 after the MDN vaccination. Similarly, 35 cases of myocarditis/pericarditis were identified, with 15 occurring after the BNT vaccination, 12 after the MDN vaccination, and 8 after the AZ vaccination. Anaphylaxis occurred in six cases, including one case after AZ vaccination, three cases after MDN vaccination, and two cases after MVC vaccination.

### 3.4. Fatal Cases

There were 920 fatal cases, including 357 cases (38.8%) in those over 80 years old, 367 cases (39.9%) between 60 and 79 years, 149 cases (16.2%) between 40 and 59 years, 39 cases (4.2%) between 21 and 39 years, and 8 cases (0.9%) in individuals less than 20 years old (Appendix A). Cardiovascular diseases were the leading cause of death (464 cases, 50.4%), followed by infectious diseases (205 cases, 22.3%), neurological diseases (102 cases, 11.3%), and hematological diseases (26 cases, 2.8%) (Appendix A). Most of the fatal cases (896 cases, 97.4%) were unassociated with vaccination, while five cases (0.5%, four females, and one male) were judged as causally associated with vaccination. All five cases occurred after the first dose of the AZ vaccine and were diagnosed with TTS. There were 19 cases (2.1%) judged as having an indeterminate association with vaccination, including 15 cases of cardiovascular diseases (heart failure [*n* = 9], myocarditis [*n* = 5], and acute myocardial infarction [*n* = 1]); 3 cases of hematological diseases (subdural hemorrhage due to thrombocytopenia [*n* = 2] and idiopathic thrombocytopenia purpura [*n* = 1]); and 1 case of anaphylactic shock.

### 3.5. Autopsy Cases

Autopsies were performed on 198 fatal cases, with none linked to vaccination. Among these, 162 cases (81.8%) had pre-existing diseases, and 95 cases (47.9%) had new, unrelated conditions identified. Cardiovascular diseases were the most common newly identified condition (38.9%), followed by neurological diseases (15.8%), pulmonary diseases (12.6%), renal disease (1.1%), and others (14.7%). In 16 autopsy cases (16.8%), deaths were classified as unnatural due to choking, trauma, intoxication, and other causes.

## 4. Discussion

Vaccination is a cornerstone in controlling the spread of severe acute respiratory syndrome coronavirus 2 and saving lives. However, like any medication, AEFI can occur. To compensate individuals who experience serious AEFI potentially linked to COVID-19 vaccines, many countries have established VICPs. Due to the rapid development of the pandemic, COVID-19 vaccines were swiftly authorized for emergency use, raising doubts about their safety [5]. Taiwan’s VICP stands out for its unique features, offering free and easy application processes and providing subsidies for autopsies in suspected vaccine-related deaths. These factors contribute to a high volume of applications, potentially overwhelming the review process. To quickly gain insight into the association between AEFI and COVID-19 vaccines, we analyzed preliminary data from Taiwan’s VICP.

The number of vaccine compensation applicants is likely correlated with the number of vaccine doses administered. This study primarily focused on cases involving early recipients, resulting in a significantly higher proportion of applications for first doses compared to subsequent doses. The AZ vaccine was the first COVID-19 vaccine authorized in Taiwan [1]. A higher number of vaccine compensation applicants in this study does not necessarily equate to a higher risk of adverse events. People are more likely to attribute health problems to vaccination if symptoms appear soon after inoculation. In this study, nearly one-third of AEFI occurred within 1 day of vaccination, and almost two-thirds occurred within 1 week.

Elderly individuals with pre-existing health conditions are more susceptible to experiencing AEFI. The presence of comorbidities further complicates the assessment of causality. Patients with conditions such as hypertension, poorly controlled diabetes, or end-stage renal disease on dialysis are more prone to sudden health deterioration, constituting a significant proportion of applicants. In Taiwan, the National Health Insurance System provides access to medical records for the past 3 years, which the VICP committee uses to review the medical histories of vaccines. However, limitations in recordkeeping may result in the underreporting of pre-existing conditions.

Acute symptoms from cardiovascular or neurological conditions can deteriorate rapidly and are more likely to be misattributed to vaccination, leading to a higher rate of compensation claims. Cardiovascular diseases accounted for approximately one-quarter, and neurological conditions for over one-fifth of the applications in this study. Infectious diseases, logically unrelated to vaccination, constituted one-eighth of reports, possibly due to postvaccination fever raising suspicion among applicants. Beyond the higher number of people initially receiving the AZ vaccine, another factor contributing to a higher rate of compensation claims from AZ vaccine recipients is likely the public awareness of possible thrombosis events after AZ vaccination, which was heavily publicized during the program’s rollout.

Standards for assessing the association between COVID-19 vaccines and AEFI vary globally [6,7]. Our findings revealed that the majority of AEFI (85.5%) was not attributed to the vaccine, suggesting a relatively high safety profile. Many cases were deemed indeterminate owing to symptom timing, clinical presentation, or insufficient supporting data. Because more population-based studies on the relationship between vaccination and AEFI have been published, these judgments may be revised.

A higher proportion of females was observed in the associated or indeterminate group, which may be linked to AZ vaccines with hematological complications. Older individuals with underlying health conditions are more likely to have non-vaccine-related reasons for AEFI, resulting in most cases being unassociated. In this study, thrombosis events were predominantly associated with the AZ vaccine, consistent with previous research [8,9]. AEFI complaints were more common after the first dose of the AZ vaccine, aligning with the existing literature [10]. Despite reports of higher AEFI rates after the second dose of mRNA vaccines [2], our study found more AEFI after the first dose of mRNA vaccines. This discrepancy may be due to our focus on the initial data from the COVID-19 vaccine compensation cases. Nonetheless, even within this limited timeframe, a trend toward a higher rate of AEFI after the second dose of the MDN vaccine compared to the AZ vaccine was observed in this study.

The most documented reactions associated with COVID-19 vaccines include thrombosis/TTS [11,12], myocarditis/pericarditis [13,14], GBS [14,15], and anaphylaxis [16]. TTS, a rare but serious adverse event linked mostly to the AZ vaccine, is characterized by low platelet counts and blood clots in large blood vessels [5]. Previous studies suggest two main mechanisms: the action of antiplatelet factor 4 antibodies and the direct interaction between the adenovirus vector and platelets [17,18]. In our study, myocarditis and pericarditis were more commonly associated with BNT and MDN vaccines, consistent with the higher proportion of cardiovascular complications in existence, including myocarditis and pericarditis, which were reported with mRNA vaccines [3,19,20]. The suspected mechanisms for COVID-19 mRNA vaccine-induced myocarditis involve hyperimmunity, potentially due to mRNA immune reactivity, the production of antibodies to combat SARS-CoV-2 spike glycoproteins cross-reacting with myocardial contractile proteins, and hormonal differences [21,22]. Neurological disorders, such as GBS or encephalitis, can be challenging to link to the vaccine [23]. Diagnosis often relies on the timing of onset and the exclusion of other diagnoses, leading to some cases being categorized as indeterminate [24]. Similarly, urticaria and certain skin allergic reactions can fall into this category [25]. Anaphylaxis is diagnosed based on clinical presentation and occurs rarely with COVID-19 vaccines, typically without lasting effects [26].

The vast majority of death cases reviewed were not linked to the COVID-19 vaccination. Older individuals with pre-existing health conditions have a higher mortality rate, and most deaths are unrelated to vaccination, often attributed to the exacerbation of the pre-existing illnesses. Autopsies have confirmed these findings and sometimes uncovered pre-existing health problems. Occasionally, autopsies may reveal unnatural causes of death. Taiwan’s VICP offers funeral subsidies to promote autopsies, which is a unique feature of the program.

Our study has several limitations. The VICP is a passive reporting system, which may result in underreporting. In Taiwan, the government encourages people to report adverse events following the COVID-19 vaccination through a free and convenient reporting system. However, retrospective medical records may have missing data, affecting the accuracy of determining the association between the vaccine and adverse events. We only analyzed approximately one-third of the total applicants, so our findings are preliminary and may not represent the full scope of the situation. As new data on COVID-19 vaccines continue to emerge, decisions made by the VICP may change based on updated evidence from the published literature.

## 5. Conclusions

This is a preliminary report, and more data and time are required to investigate additional cases before drawing a definitive conclusion. There are new variants of SARS-CoV-2, and the vaccine strains that might cause different reactions could be changed. However, based on the current information from Taiwan’s VICP, severe adverse reactions associated with COVID-19 vaccines are rare. Therefore, the COVID-19 vaccination can be considered safe. The VICP is currently overwhelmed by a large volume of applications, which cannot be resolved in a short timeframe. It may be advisable to consider implementing more stringent regulations for compensation applications, especially for cases that are clearly unrelated.

## Figures and Tables

**Figure 1 healthcare-12-01250-f001:**
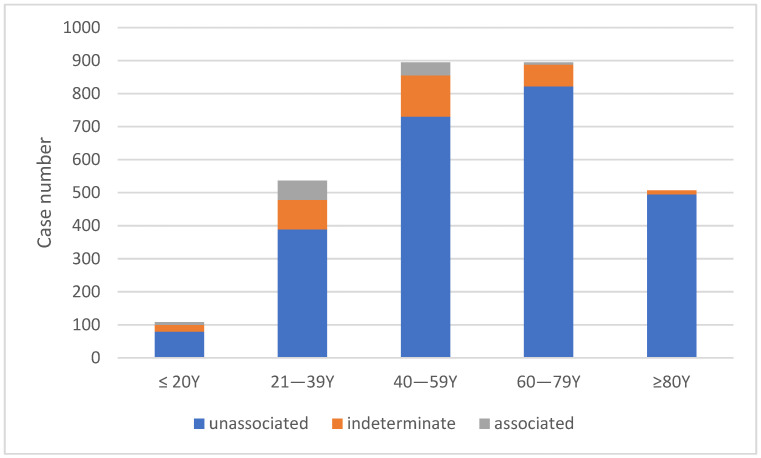
Age distribution of applicants and their association with COVID-19 vaccines.

**Figure 2 healthcare-12-01250-f002:**
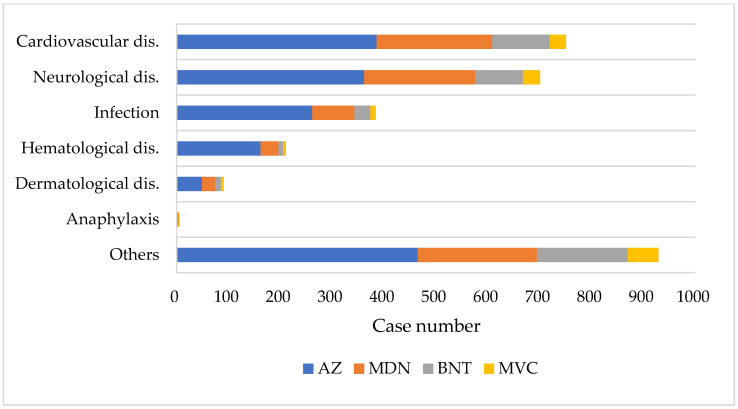
Diagnoses associated with compensation claims by vaccine type. Abbreviations: disease (dis.), Oxford/AstraZeneca (AZ), Moderna (MDN), Pfizer/BioNTech (BNT), and Medigen (MVC).

**Figure 3 healthcare-12-01250-f003:**
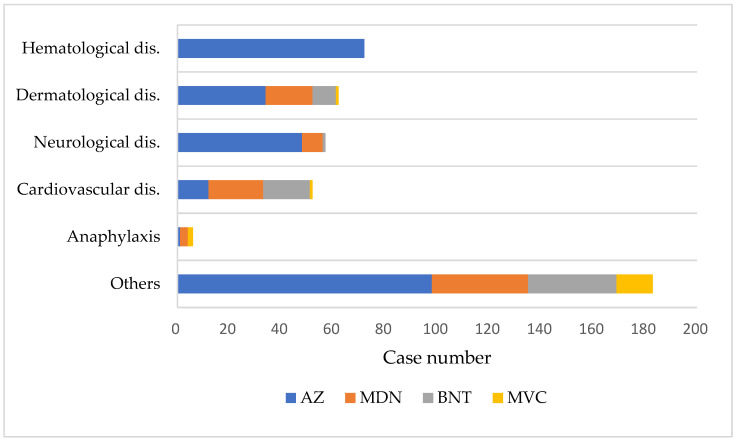
Diagnoses of Adverse Events Following Immunization in combined associated/indeterminate applicants by vaccine type. Abbreviations: disease (dis.), Oxford/AstraZeneca (AZ), Moderna (MDN), Pfizer/BioNTech (BNT), and Medigen (MVC).

**Table 1 healthcare-12-01250-t001:** Comparison between associated/indeterminate and unassociated groups.

	Associated/ Indeterminate *n* = 426	Unassociated *n* = 2515	
	Case Number (%)	Case Number (%)	*p*
Female	238 (55.9)	1156 (46.0)	<0.05
Age (years)	40.7 ± 21.8	58.2 ± 22.5	<0.05
Pre-existing chronic disease *	209 (66.1)	1477 (78.3)	<0.05
Symptom onset time interval (days)	4.5 ± 7.2	5.4 ± 11.3	0.129
Death	11 (2.6)	324 (12.9)	<0.05
Vaccine			
AZ	259 (60.8)	1331 (52.9)	<0.05
MDN	82 (19.2)	700 (27.8)	<0.05
BNT	65 (15.3)	361 (14.4)	0.603
MVC	20 (4.7)	123 (4.9)	0.711

* Among 2202 cases with complete medical records logged in Vaccine Injury Compensation Program data.

## Data Availability

The datasets used and/or analyzed during the current study are available from the corresponding author upon reasonable request.

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
