# Peer review of "Preliminary Report of Nationwide COVID-19 Vaccine Compensation in Taiwan"

_healthcare, 2024, doi:10.3390/healthcare12131250_

Round 1
Reviewer 1 Report
Comments and Suggestions for Authors
Dear Authors,
thank you for your hard work Could you kindly incorporate the following adjustments;
in line,102, could you please include the abbreviation for AEFI
In Figure 1, could you provide more detailed explanations for the terms intermediate, associated, and unassociated
line 128, it was mentioned that the unassociated group exhibits a higher MCV vaccination rate and lower AZ (this statement should not be construed as applying solely within the group). Rather, it should be viewed in the context of the comparison between the unassociated, intermediate, and associated.
Reviewer 2 Report
Comments and Suggestions for Authors
Overall the author(s) explained the things clearly and data presentation is fine. I observed some grammatical mistakes which need to be corrected. The file with highlighted text is attached with this report.
Kindly incorporate the highlighted mistakes.

The file with highlighted text that needs correction is attached.
Author Response
Thank you for the comment. We have made changes in the manuscript accordingly.
Proof of English editing is provided separately.

Reviewer 3 Report
Comments and Suggestions for Authors
The authors have nicely presented the data for vaccine compensation claims in Taiwan. But I could not understand the scientific significance of this data. The complications and co-morbidities are listed only for the patients who applied for compensation. It will be great if you can perform same analysis for people who have not applied for compensation.
Comments on the Quality of English LanguageThe language used for manuscript is easy to understand but can be improved to convey the finding in a clearer way.
Author Response
Reviewer #3:
The authors have nicely presented the data for vaccine compensation claims in Taiwan. But I could not understand the scientific significance of this data. The complications and co-morbidities are listed only for the patients who applied for compensation. It will be great if you can perform same analysis for people who have not applied for compensation.
Response: Thanks for your suggestion. However, it is important to note that VICP is a passive surveillance system. Unfortunately, we did not have access to comorbidity data for non-VICP cases. Nevertheless, the high ratio (76.6%) of pre-existing chronic diseases among applicants was notably higher than that of the general population.
Proof of English editing is provided, please see the attachment.

Reviewer 4 Report
Comments and Suggestions for Authors
This study investigated adverse effect of covid 19 vaccination which decided by VICP of Taiwan and found that most adverse effect not casually linked to vaccination.
How VICP determined and classified to causally associated, indetermined and no causal association, good to be explained. Especially causally associated and indetermined should be clearly differentiated.
In table 1 the author compared the type of vaccine and the adverse effect and appeared that ZA type vaccine significantly associated/indeterminate and MDN type vaccine significantly unassociated. It is good if the author adjusted to pre-existing diseases, age and sex because those factors also statistically significant – before comparing of these vaccine type.
In Figure S5 and S6 it is interesting to add the mark of vaccine type, as presented in another figures.
Author Response
Please see the attachment. Proof of English editing is provided separately.

Reviewer 5 Report
Comments and Suggestions for Authors
Dear Authors,
Thank you for your manuscript entitled " Preliminary Report of Nationwide COVID-19 Vaccine Compensation in Taiwan"
I have some comments that I wish it may help :
1- you have studied the clinical complications based on the type of vaccine.
are these complications comes out from single vaccination or 2 shots of the same vaccine or 3 shots?
2- the duration of the study is very long which is intervened with many viral variants as well as developed generations of vaccine, please check for this topoc.
3- Do do you confirm that the clinical complications are not coexist because of other factors like genetic predisposition or chronic disease or even co-infection with different viral variant than what is supposed to be protected from by the vaccine ?
Thank you
Reviewer 6 Report
Comments and Suggestions for Authors
To: Healthcare MDPI
Dear EIC,
Dear AE,
This is my review results for the manuscript ID: healthcare-3005862
This is a preliminary report from Taiwan’s VICP. I think this manuscript can add some novel things to the field by clarifying the dark aspects of the COVID-19 vaccination side effects. The manuscript is well-written, and data analysis is simple and acceptable for all readers and researchers from various research fields. I focused on the manuscript several times. Totally, writing and presentation is acceptable; however, I find some comments described below.
Comments
· The manuscript should be supplied with a Table consisting of all data, including patients’ characteristics (gender, age group, vaccine name, underlying diseases, etc.). This table shall be located in the manuscript body (not supplementary data). This is a major issue.
· Kindly describe the full name of each vaccine in the legend of Figure S1, Figure 2, and Figure 3.
Round 2
Reviewer 5 Report
Comments and Suggestions for Authors
Dear Authors,
Thank you for your corrections.
I have a simple comment, please make your clinical data more detailed and clearer in the future. This will be a scientific evidence where others can get benefit of it.
Good luck
Reviewer 6 Report
Comments and Suggestions for Authors
Dear EIC,
Dear AE,
This is my review results for the revised version of manuscript ID: healthcare-3005862.
I think the manuscript qualified for further publishing processes in the journal.